# Prevalence of mastitis and its associated risk factors in lactating camels in Northern Egypt

Abdelfattah Selim●[1]*, Mohamed Marzok[2]*, Hattan S. Gattan[3,4], Hesham Ismail[5]

1 Department of Animal Medicine (Infectious Diseases), Faculty of Veterinary Medicine, Benha University, Toukh, Egypt, 2 Department of Microbiology, College of Veterinary Medicine, King Faisal University, Al-Asha, Saudi Arabia, 3 Department of Medical Laboratory Sciences, Faculty of Applied Medical Sciences, King Abdulaziz University, Jeddah, Saudi Arabia, 4 Special Infectious Agents Unit, King Fahad Medical Research Center, King AbdulAziz University, Jeddah, Saudi Arabia, 5 Department of Public Health, College of Veterinary Medicine, King Faisal University, Hofuf, Saudi Arabia

* Abdelfattah.selim@fvtm.bu.edu.eg (AS); mmarzok@kfu.edu.sa (MM)

## Abstract

Dromedary camels are susceptible to mastitis, a multifactorial disease affecting dairy animals worldwide and leading to significant economic losses, particularly due to its subclinical form. However, limited data exist on the prevalence and risk factors associated with lactating camel mastitis in Egypt. Therefore, a cross-sectional study was conducted across three Egyptian governorates to assess the prevalence and identify potential risk factors. A total of 390 lactating camels were examined for both clinical and subclinical mastitis using the California Mastitis Test (CMT). The overall prevalence of mastitis was 38.46% (150/390), comprising 6.4% (25/390) clinical and 32.1% (125/390) subclinical cases. Significant associations ($p < 0.05$) were observed between mastitis prevalence and factors such as age, lactation stage, tick infestation, milking hygiene, and the presence of udder or teat lesions. The likelihood of mastitis was six times higher in camels older than five years, three times higher during early lactation, twice as high in the presence of tick infestation, twice as high with poor milking hygiene, and three times higher in animals with udder or teat lesions. Among the 600 quarter milk samples obtained from positive animals, mastitis-causing pathogens were isolated from 380 samples (63.3%), while 220 samples (36.7%) showed no bacterial growth. *Streptococcus* spp. (excluding *S. agalactiae*) and *Escherichia coli* were the most prevalent isolates (26.1% and 25%, respectively), whereas *S. agalactiae* was the least frequent, detected in only 5.5% of the samples. These findings highlight the importance of implementing integrated control measures to reduce mastitis prevalence, enhance camel milk quality, mitigate economic losses, and safeguard public health.

**Data availability statement:** All data available in the manuscript.

**Funding:** This work was supported through the Annual Funding track by the Deanship of Scientific Research, Vice Presidency for Graduate Studies and Scientific Research, King Faisal University, Saudi Arabia (Grant KFU252782 to MM).

**Competing interests:** The authors have declared that no competing interests exist.

# 1. Introduction

The camel is a versatile animal that performs very well in arid and semi-arid conditions with little forage and water [1]. It plays a significant role in human survival and utilization of these arid and dry regions. The growing human population in developing nations has increased the focus on using formerly underutilized animals to provide milk and meat. However, in arid regions affected by extreme heat, water scarcity, and limited feed availability, camels serve as a more reliable food source than cows [2]. Camel milk is rich in minerals, protein, fat, and vitamins, especially vitamin C. In many aspects, camel milk surpasses the milk of other domestic animals. Camel milk is also recognized for its therapeutic properties, which have been used to treat a variety of maladies [3–5].

Mastitis is an inflammatory disease that impacts the udder, leading to pathological tissue changes and alterations in the chemical and physical properties of milk. It can also cause damage to the mammary glands of camels [6–8]. The primary method of transmission for udder infections causing mastitis in camels is through the teat canal, where pathogens can enter from the environment or from the udders of infected animals and then reach the mammary gland during milking [9]. Mastitis can be classified into two types: subclinical, where symptoms are not visible to workers, and clinical, where symptoms are easily detectable. In camel herds managed by pastoralists, both types, including subclinical mastitis (SCM), are commonly observed and often linked to intramammary infections [10,11]. According to Seligsohn, Younan [12], these conditions can negatively impact household finances and pose significant health risks to both humans and animals.

Clinical mastitis is characterized by a swollen, discolored udder that is painful to palpation, along with changes in the color and quality of the milk, which vary depending on the level of inflammation [13,14]. In addition, increased somatic cell counts and changes in the levels of salt, potassium, magnesium calcium and lactose, are significant chemical alterations in milk. Consuming contaminated milk can lead to human milk-borne diseases, presenting a serious public health risk [6,15]. Research indicates that subclinical mastitis has a significant financial impact and has a greater influence on lactating animal productivity than clinical cases [11,16].

The severity and spread of this common and widespread disease are influenced by numerous risk factors, including milking practices, milk production level, hygiene, age, breed, parity, and stage of lactation [17–19].

Camel mastitis has been reported in a number of camel-rearing nations, such as Sudan [20], Kenya [21], Somalia [22,23], Algeria [17], Saudia Arabia [24], United Arab Emirates [25] and Egypt [10,26]. Previous studies on camel mastitis have primarily focused on its etiology, epidemiology, and treatment. However, limited information is available regarding the risk factors associated with this disease in Egypt. Gaining insight into these factors is essential for developing effective disease management strategies aimed at reducing both morbidity and mortality.

Consequently, this study aimed to determine the prevalence and significant risk factors for camel mastitis.

## 2. Materials and methods

### 2.1. Ethical statement

All methods and procedures in this study were conducted in accordance with the guidelines and approval of the Ethics Committee of the Faculty of Veterinary Medicine, Benha University. Verbal consent was obtained from the animal owners prior to the collection of milk samples and documented by form and reviewed by Ethics Committee of the Faculty of Veterinary Medicine, Benha University.

### 2.2. Study area

The study was conducted in three Egyptian governorates: Kafr El-Sheikh, Qalyubia, and Menofia, all of which are located in northern Egypt's Nile Delta region, which is known for its heavy agricultural activity and large population of domesticated animals, including camels.

Kafr El-Sheikh is located in the northernmost region of the Delta, near the Mediterranean Sea. It is primarily rural, noted for its fertile soils and vast agricultural lands. The governorate has a Mediterranean climate, with warm rainy winters and scorching, dry summers. Qalyubia is located just north of Cairo and is part of the Greater Cairo Metropolitan Area. It acts as a transitional zone between urban and agricultural environments, taking use of its location along the Nile River. Qalyubia's climate is semi-arid, with hot summers and mild winters. Rainfall is rare. Menofia, located in the central Nile Delta, is primarily rural and surrounded by other agricultural governorates. It is defined by fertile plains and a moderate climate characterized by high humidity, scorching summers, and chilly winters. These three governorates were chosen for the study because they play an important role in livestock production and are home to traditional animal husbandry traditions.

### 2.3. Sample size and sampling

The sample size of animals was determined using the formula provided by Thrusfield [27], based on an assumed minimum expected prevalence of 50%, a desired absolute precision of 5%, and a 95% confidence level.

$$N = 1.96^{2Pexp(1-Pexp)}/d^2$$

Where $n$ represents the required sample size, 1.96 is the Z-value corresponding to the 95% confidence level, $Pexp$ denotes the expected prevalence (50%), and $d$ indicates the desired absolute precision (0.05). A total of 390 animals were examined, representing 1560 udder quarters

### 2.4. Milk sampling and examination

Each animal was individually identified, and a clinical examination of the udder was conducted through visual inspection and palpation. The udder was palpated and examined visually for lesions, symmetry, size, and clinical indications of mastitis. The milk's quality, color changes, and presence of any obviously aberrant ingredients were also assessed. Clinical mastitis was defined as an udder quarter with visible inflammatory changes in the mammary gland tissue such as redness, swelling, pain, or increased heat and/or visible alterations in the milk, such as changes in color (e.g., watery, bloody, blood-tinged, or serum-like) or consistency (e.g., presence of clots, flakes, stringy, or viscous texture) [28].

Depending on the owner's milking schedule, a single sample was taken from each camel, either in the early morning or late afternoon. Prior to sampling, the teats were disinfected using cotton soaked in 70% alcohol. Milk samples were then aseptically collected, inspected for any visible abnormalities, and screened for mastitis using the California Mastitis Test (CMT), following the method described by Quinn [29].

Milk was squirted into the appropriate cups of the CMT paddle from each udder quarter. After that, an equal volume of CMT reagent was added to each cup and was carefully mixed. The results were interpreted as follows: a CMT score of

0 was considered negative, whereas scores of trace, 1+, 2+, and 3+were regarded as positive, resulting in five distinct categorical classifications.

## 2.5.  Questionnaire-based information collection

The questionnaire was created to collect data on various aspects of the study animals, including locality, age (≤3, 3–5 and > 5 years), body condition (good, medium and poor), lactation stage, tick infestation status, hygiene of milk process, and the presence of udder or teat lesions. The lactation stage was classified into three categories—early (1–2 months), middle (3–9 months), and late (10–18 months)—to see whether there is a substantial difference in the incidence of mastitis between these times. Camels in good condition have a full hump, smooth body contours, and well-covered ribs and pelvis; those in medium condition show partial hump reduction, slightly visible ribs, and mildly prominent bones; while camels in poor condition have a sunken hump, clearly visible ribs and spine, and severe muscle and fat loss. Additionally, the interview aimed to assess the extent of tick infestation, the implementation of hygienic practices during the milking process, and the presence of udder or teat lesions.

## 2.6.  Bacterial isolation and identification

Milk samples that tested positive using the CMT were examined bacteriologically. For bacteriological analysis, 10 μL of milk from each sample was inoculated onto sheep blood agar and MacConkey agar (Oxoid Ltd., Cambridge, UK). The plates were incubated aerobically at 37°C and examined for bacterial growth after 24 and 48 hours. Presumptive identification of bacterial isolates was based on colony morphology, Gram staining, hemolysis patterns, catalase activity, potassium hydroxide (KOH) reaction, and additional biochemical tests [30]. A pure culture yielding ≥5 colony-forming units (CFU) was considered significant, except for *Streptococcus agalactiae* and *Staphylococcus aureus*, which were deemed significant if ≥1 CFU was detected.

Bacterial isolates were subsequently subcultured into their respective selective media for further characterization and species identification. Gram-positive cocci were initially tested for catalase activity; catalase-positive isolates underwent coagulase testing. *Staphylococcus* species were identified based on their growth on mannitol salt agar and results from coagulase, catalase, and oxidase tests. *S. aureus* was distinguished from other *Staphylococcus* spp. by a positive coagulase reaction and maltose fermentation. *Streptococcus* isolates were characterized using the CAMP test, esculin and sodium hippurate hydrolysis, catalase activity, and sugar fermentation profiles. Specifically, *S. agalactiae* was differentiated from other mastitis-associated streptococci by its CAMP reaction, esculin hydrolysis on Edwards medium, and ability to grow on MacConkey agar. Gram-negative isolates were further identified using a series of biochemical tests, including triple sugar iron (TSI), motility, urease activity, and oxidase reaction.

## 2.7.  Data analysis

The collected data were entered into a Microsoft Excel spreadsheet for statistical analysis. The prevalence of mastitis in lactating camels was analyzed across various categorical variables, including locality, age, body condition, lactation stage, presence of tick infestation, hygiene practices during the milking process, and the occurrence of udder or teat lesions. Furthermore, the CMT result was treated as the dependent variable, and the impact of selected risk factors on this outcome was analyzed using univariate and multivariable logistic regression in SPSS software (IBM, USA). The odds ratio (OR) and its 95% confidence interval (CI) were computed as an effect measure. A statistically significant p-value was defined as less than 0.05. The model's fit was assessed using the Hosmer-Lemeshow goodness-of-fit test.

## 3.  Results

A total of 390 lactating camels were examined for both clinical and subclinical mastitis, resulting in an overall prevalence of 38.46% and 28.8% on animal wise and quarter wise, respectively. Of these, 6.4% were affected by clinical mastitis and 32.1% by subclinical mastitis, as shown in Table 1.

**Table 1. Prevalence of clinical and subclinical mastitis in lactating camels.**

| Form of mastitis | No of positive animal[a] | Camel- level Prevalence % | No of positive quarter[b] | Quarter-level prevalence% |
|---|---|---|---|---|
| Clinical | 25 | 6.4 | 190 | 12.2 |
| Subclinical | 125 | 32.1 | 260 | 16.7 |
| Total | 150 | 38.46 | 450 | 28.8 |

[a]The total number of examined animals 390

[b]The total number of examined quarter 1560

The results revealed that mastitis in camels was not significantly (P>0.05) associated with locality or the body condition of the animals. The highest prevalence was observed in Menofia governorate (38.46%) and among animals in good condition (42.61%), Table 2.

Chi-square analysis showed that age, lactation stage, tick infestation, hygiene practices during the milking process, and the presence of lesions on the udder or teat were significantly associated (p<0.05) with the prevalence of mastitis in lactating camels among the potential risk factors considered in the study, as presented in Table 2.

**Table 2. Prevalence of mastitis in camels in relation to risk factors.**

| Variable | No of examined camels | No of positive | % | 95%CI | Statistic |
|---|---|---|---|---|---|
| **locality** | | | | | |
| Qalyubia | 135 | 56 | 41.48 | 37.04-53.6 | $\chi2=1.082$ P=0.582 |
| Kafr ElSheikh | 125 | 44 | 35.20 | 23.74-39.78 | |
| Menofia | 130 | 50 | 38.46 | 30.54-47.04 | |
| **Age** | | | | | |
| ≤3 | 40 | 8 | 20.00 | 10.5-34.76 | $\chi2=19.618$ P<0.0001* |
| >3-5 | 210 | 69 | 32.86 | 26.86-39.47 | |
| >5 | 140 | 73 | 52.14 | 43.92-60.25 | |
| **Boody condition** | | | | | |
| Good | 115 | 49 | 42.61 | 33.95-51.74 | $\chi2=1.337$ P=0.513 |
| Medium | 121 | 46 | 38.02 | 29.87-46.91 | |
| Poor | 154 | 55 | 35.71 | 28.57-43.54 | |
| **Lactation stage** | | | | | |
| Early | 132 | 66 | 50.00 | 41.59-58.41 | $\chi2=13.272$ P=0.001* |
| Medium | 143 | 41 | 28.67 | 21.89-36.56 | |
| Late | 115 | 43 | 37.39 | 29.09-46.51 | |
| **Tick infestation** | | | | | |
| Yes | 143 | 64 | 44.76 | 36.85-52.94 | $\chi2=13.136$ P<0.0001* |
| No | 247 | 86 | 34.82 | 29.15-40.95 | |
| **Hygiene of milk process** | 390 | 150 | | | |
| Good | 127 | 34 | 26.77 | 19.83-35.07 | $\chi2=10.873$ P=0.001 |
| Poor | 263 | 116 | 44.11 | 38.24-50.15 | |
| **Lesion on udder or teat** | | | | | |
| Yes | 112 | 64 | 57.14 | 47.89-65.92 | $\chi2=23.168$ P<0.0001* |
| No | 278 | 86 | 30.94 | 25.8-36.6 | |
| Total | 390 | 150 | 38.46 | 33.77-43.38 | |

*Results are significant at P value<0.05

The prevalence rate was higher in camels over 5 years old (52.14%), in those at the early lactation stage (50%), in cases with tick infestation (44.76%), in animals with poor hygiene during the milking process (44.11%), and in camels with lesions on the udder or teat (57.14%), Table 2.

Similarly, a higher likelihood of mastitis prevalence was observed in camels over 5 years of age (P < 0.0001, OR=6.2), during the early lactation stage (P = 0.001, OR=2.5), in the presence of tick infestation (P = 0.011, OR=1.5), in cases where milking hygiene was not practiced (P = 0.001, OR=2.4), and among camels with udder or teat lesions (P < 0.0001, OR=2.5), Table 3.

Among the 600 quarter milk samples representing the positive animals were analyzed bacteriologically, mastitis caus-ing pathogens both Gram-positive and Gram-negative were detected in 380 samples (63.3%), while 220 samples (36.7%) showed no bacterial growth. Among the isolates, *Streptococcus* species (excluding *S. agalactiae*) and *Escherichia coli* were the most prevalent, accounting for 26.1% and 25% respectively, whereas *S. agalactiae* was the least frequently iden-tified, representing only 5.5%, as shown in Table 4.

## 4. Discussion

Mastitis is a considerable impediment to milk production in *Camelus dromedarius* in Egypt's dry and semi-arid regions. According to several reports, mastitis is on the rise among traditionally maintained camels and is expected to rise further as milk production per individual camel improves.

**Table 3. Multivariable logistic regression analysis for risk factors associated with prevalence of mastitis in camels.**

| Variable | B | S.E. | OR | 95% CI for OR | | P value |
|---|---|---|---|---|---|---|
| | | | | Lower | Upper | |
| **Age** | | | | | | |
| >3-5 | 1.152 | 0.459 | 3.2 | 1.29 | 7.79 | 0.012 |
| >5 | 1.830 | 0.464 | 6.2 | 2.51 | 15.48 | <0.0001 |
| **Lactation stage** | | | | | | |
| Early | 0.897 | 0.278 | 2.5 | 1.42 | 4.23 | 0.001 |
| Late | 0.168 | 0.300 | 1.2 | 0.66 | 2.13 | 0.575 |
| **Tick infestation** | | | | | | |
| Yes | 0.380 | 0.241 | 1.5 | 0.91 | 2.34 | 0.011 |
| **Hygiene of milk process** | | | | | | |
| Poor | 0.881 | 0.265 | 2.4 | 1.43 | 4.06 | 0.001 |
| **Lesion on udder or teat** | | | | | | |
| Yes | 0.899 | 0.251 | 2.5 | 1.50 | 4.02 | <0.0001 |

B: Logistic regression coefficient, SE: Standard error, OR: Odds ratio, CI: Confidence interval

**Table 4. Bacterial species isolated from quarter milk samples from lactating camels.**

| Bacterial isolate | Number of isolates (clinical/ Subclinical) | % of bacterial isolates |
|---|---|---|
| *Escherichia coli* | 100 (25/75) | 25 |
| *S. aureus* | 85 (36/49) | 22.4 |
| Corynebacteria spp. | 49 (18/31) | 12.9 |
| Coagulase negative Staphylococci | 99 (29/70) | 26.1 |
| *Micrococcus* spp. | 31 (12/19) | 8.2 |
| *S. agalactiae* | 21 (6/15) | 5.5 |
| Total | 380 (126/254) | 100 |

The overall mastitis prevalence of 38.46% observed in this study aligns with a previously reported rate of 34.7% in the Borena Zone of Oromia Regional State, Ethiopia [31]. However, it is lower than the prevalence reported in the Afar Region of Ethiopia (59.8%) [32], selected pastoral areas of eastern Ethiopia (76%) [33], and the Yabello district of Borena Zone (44.8%) [34]. Conversely, it is higher than the prevalence rates reported in Abu Dhabi, United Arab Emirates (18.5%) [35], Jijiga town in eastern Ethiopia (30.2%) [13], and Gursum district of the Hararghe Zone (31%) [35].

The variation in mastitis prevalence rates in camels between different countries can be attributed to several key factors. Differences in management practices play a significant role; in regions where camels are raised under traditional pastoral systems with limited veterinary care and poor milking hygiene, the prevalence tends to be higher [36]. Access to veterinary services and disease surveillance also influences the prevalence, as regular monitoring allows for early detection and treatment of mastitis [37]. Additionally, environmental conditions, such as dusty or unsanitary settings, and differences in climate can further impact disease occurrence. Genetic factors, such as breed susceptibility, and methodological differences in study design and diagnostic techniques also contribute to the disparities in reported prevalence rates across countries [11,20,35,38–40].

In addition, the percentages of clinical (6.4%) and subclinical (32.1%) mastitis found in this study are consistent with those of Megersa [41], who found that the prevalence of subclinical mastitis in dromedary camels in the Borana region of Southern Ethiopia ranged from 28.6% to 37.6%, while the prevalence of clinical mastitis ranged from 10% to 17%.

The clinical mastitis findings in this study are similar with the 5.9% prevalence reported by Abdurahman, Agab [42] in Sudan and the 8.3% found by Abera, Abdi [43] in Jijiga. Woubit, Bayleyegn [44] reported a slightly lower frequency of 2.1%. In contrast, Barbour, Nabbut [45] in Saudi Arabia, Megersa [41] in Borana, Southern Ethiopia, and Obied, Bagadi [46] in Sudan all reported higher clinical mastitis rates of 15%, 17%, and 19.5%, respectively.

In most studies, including the current one, the prevalence of clinical mastitis is lower than that of subclinical mastitis [47–49]. This difference is likely because subclinical mastitis frequently remains undetected, as affected animals usually exhibit no obvious clinical signs and continue to produce milk that appears normal, making diagnosis more challenging without routine testing [8,50–52].

The CMT results revealed a strong association between age and the prevalence of mastitis. This finding is consistent with the findings of Zeryehun, Haro [53] and Aqib, Ijaz [54], who found a statistically significant (p < 0.05) association between adult age (≥5 years) and the occurrence of mastitis in lactating camels in the Borena lowlands in southwestern Ethiopia and the Cholistan Desert of Pakistan.

The effect of age on the prevalence of mastitis in camels can be attributed to several physiological and management-related factors. As camels age, they are likely to have undergone multiple lactation cycles, which increases their exposure to potential pathogens and environmental risk factors over time. Repeated milking and previous udder infections may lead to structural and functional changes in the mammary gland, such as fibrosis or reduced teat canal integrity, making older camels more susceptible to new infections. Additionally, older animals may experience a gradual decline in immune function, which can compromise their ability to effectively combat infections [8,11,25,55,56].

In the current study, the body condition score of the animals was not significantly associated (p > 0.05) with the prevalence of mastitis in lactating camels. In contrast, studies by Zeryehun, Haro [53], Aqib, Ijaz [54] found a significant association between poor (thin) body condition and the occurrence of mastitis in dromedary camels in the Cholistan Desert of Pakistan and the pastoral areas of the Borena lowlands in Southern Ethiopia,. As a result, a more thorough examination is required to determine the underlying causes of this disparity.

The early lactation stage showed a significant association with the prevalence of mastitis in camels, which is consistent with the findings of Regassa, Golicha [34]. This might be due to most new infections, especially those caused by environmental pathogens, tend to happen during the early dry period and the first two months of lactation [57–60]

In contrast, the presence of ticks was evaluated as a potential risk factor for camel mastitis in the study area. Previous studies has suggested that tick infestation can predispose the udder to mastitis by allowing pathogens to enter, where ticks frequently attach to the udder, causing skin and teat sores [34,61].

The findings revealed that camels subjected to poor hygiene methods throughout the milking process demonstrated a higher prevalence of mastitis, which is consistent with the findings of Ahmad, Yaqoob [14]. Similar findings were observed in a prior study on bovine mastitis in Ethiopia [62], which indicated poor milking hygiene as a major risk factor. This could be due to a lack of udder washing before milking, the employment of common milkers who may have cuts or chapped hands, and the sharing of udder cloths across animals—all of which can serve as vectors for the transfer of infectious mastitis-causing microorganisms [6,14,63–65].

According to this study, udder or teat lesions were found to be a substantial risk factor for mastitis development. Because they allow bacteria to enter and can result in long-lasting tissue damage, these lesions are one of several predisposing factors that make camels more susceptible to mastitis [61].

This study's findings are consistent with earlier studies that found camels with udder lesions had a higher prevalence of mastitis. For instance, Woubit, Bayleyegn [44] found a similar incidence of mastitis in the Borana lowlands of Southern Ethiopia, whereas Bekele and Molla [32] reported a 72.2% prevalence among infected camels in the Afar region. This result was further corroborated by Abdurahman, Agab [42], who explained that trauma can directly cause mastitis by increasing the udder's susceptibility to bacterial invasion.

The predominant bacterial genera isolated in this study—*Staphylococcus*, *Streptococcus*, *Corynebacterium*, and *Escherichia*—are consistent with previous findings by Kalla, Butswat [66] and Saleh and Faye [67], who also identified *Staphylococcus*, *Streptococcus*, and *Escherichia* as key mastitis pathogens. The isolation rate of *E. coli* in the current study aligns with the observations reported by Mengistu, Molla [68]. Since coliforms often indicate poor hygienic conditions and, to a lesser extent, fecal contamination [69], their prevalence may vary significantly depending on the level of hygiene practiced.

The prevalence of *S. aureus* (22.4%) observed in this study is significantly higher than the 0.6% reported by Almaw and Molla [70]. This discrepancy may be attributed to cultural taboos against heating camel milk and the common practice of storing milk at high ambient temperatures after milking and during transport, which creates favorable conditions for the production of staphylococcal enterotoxins, as noted by Alebie, Molla [47]. Additionally, the proportion of coagulase-negative *Staphylococci* (CNS) isolated from CMT-positive milk samples (26.1%) was also higher than the rates reported by Alebie, Molla [47] (19.57%).

The relatively low prevalence of *S. agalactiae* (5.5%) observed in the current study aligns with the findings of Husein, Haftu [13], who reported a similar rate of 3.5% in Jijiga town, Ethiopia. However, it is markedly lower than the prevalence rates reported by Seligsohn, Younan [12] (72% in Isiolo, Kenya) and Al-Tofaily and Alrodhan [71] (9.52% in parts of the middle Euphrates, Iraq). This low detection rate of *S. agalactiae* may be attributed to the treatment of mastitis cases using a combination of traditional folk remedies and modern antimicrobial therapies in the study area.

Moreover, the prevalence of *Corynebacterium* spp. (12.9%) observed in this study is notably higher than the 3.03% reported by Alamin, Alqurashi [72] in North Kordofan State, Sudan. The presence of various bacterial species in the current findings may be attributed to inadequate milking hygiene practices in the study area, particularly the lack of udder and teat washing before milking.

## Conclusions

Mastitis was detected among the examined lactating camels, with subclinical cases being more prevalent than clinical ones. While locality and body condition appeared to have an influence, they were not significantly associated with mastitis prevalence. In contrast, factors such as age, stage of lactation, tick infestation, hygiene during the milking process, and the presence of udder or teat lesions showed a strong and statistically significant association with the occurrence of mastitis. *E. coli* (25%) and coagulase-negative *Staphylococci* (26.1%) were among the predominant bacterial isolates identified, while *Streptococcus agalactiae* was the least frequently detected in this study. The bacterial species isolated from camel milk samples are known to be associated with both contagious and environmental forms of mastitis. Therefore,

implementing proper and hygienic milking practices is crucial for the effective prevention of both types of mastitis. These findings highlight the importance of improving management and hygiene practices to reduce the burden of mastitis, particularly in its subclinical form.

## Supporting information

**S1 File. Inclusivity in global researchquestionnaire.**
(DOCX)

## Author contributions

**Conceptualization:** Abdelfattah Selim, Mohamed Marzok, Hattan S. Gattan.

**Data curation:** Abdelfattah Selim, Hattan S. Gattan, Hesham Ismail.

**Formal analysis:** Mohamed Marzok, Hattan S. Gattan, Hesham Ismail.

**Funding acquisition:** Abdelfattah Selim, Mohamed Marzok, Hesham Ismail.

**Investigation:** Mohamed Marzok.

**Methodology:** Abdelfattah Selim, Mohamed Marzok, Hesham Ismail.

**Project administration:** Abdelfattah Selim.

**Resources:** Hattan S. Gattan.

**Software:** Mohamed Marzok.

**Supervision:** Hesham Ismail.

**Validation:** Hattan S. Gattan.

**Visualization:** Abdelfattah Selim, Hesham Ismail.

**Writing – original draft:** Abdelfattah Selim, Mohamed Marzok, Hattan S. Gattan, Hesham Ismail.

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
