## [Decision Letter · Decision Letter 0]

29 Jul 2025

Dear Dr. Selim,

Thank you for submitting your manuscript to PLOS ONE. After careful consideration, we feel that it has merit but does not fully meet PLOS ONE’s publication criteria as it currently stands. Therefore, we invite you to submit a revised version of the manuscript that addresses the points raised during the review process.

Although both reviewers found interest in your study, they raised a number of concerns that need to be adressed.

In particular, both reviewers asked for more details regarding reporting of prevalence of clinical and subclinical mastitis at both the camel level and the quarter level separately.

We look forward to receiving your revised manuscript.

Kind regards,

Pierre Germon

Academic Editor

PLOS ONE

Journal Requirements:

3. Please include a complete copy of PLOS’ questionnaire on inclusivity in global research in your revised manuscript. Our policy for research in this area aims to improve transparency in the reporting of research performed outside of researchers’ own country or community. The policy applies to researchers who have travelled to a different country to conduct research, research with Indigenous populations or their lands, and research on cultural artefacts. The questionnaire can also be requested at the journal’s discretion for any other submissions, even if these conditions are not met.  Please find more information on the policy and a link to download a blank copy of the questionnaire here: https://journals.plos.org/plosone/s/best-practices-in-research-reporting. Please upload a completed version of your questionnaire as Supporting Information when you resubmit your manuscript.

4. In the online submission form, you indicated that [data available from corresponding author on request].

Reviewers' comments:

Reviewer's Responses to Questions

**Comments to the Author**

1. Is the manuscript technically sound, and do the data support the conclusions?

Reviewer #1: Yes

Reviewer #2: No

2. Has the statistical analysis been performed appropriately and rigorously?

Reviewer #1: Yes

Reviewer #2: Yes

3. Have the authors made all data underlying the findings in their manuscript fully available?

Reviewer #1: Yes

Reviewer #2: Yes

4. Is the manuscript presented in an intelligible fashion and written in standard English?

Reviewer #1: Yes

Reviewer #2: Yes

Reviewer #1: This research article, titled "Prevalence of Mastitis and Its Associated Risk Factors in Lactating Camels in Northern Egypt," published in PLOS ONE, presents a cross-sectional epidemiological investigation into camel mastitis across three governorates: Kafr El-Sheikh, Qalyubia, and Menofia. The study assessed 390 lactating camels using the California Mastitis Test (CMT) and identified both clinical and subclinical cases. The overall prevalence was 38.46%, with clinical mastitis accounting for 6.4% and subclinical mastitis for 32.1% of the examined animals. Multivariable logistic regression identified several significant risk factors: camels older than five years were 6.2 times more likely to be affected; early lactation stage increased risk by 2.5 times; tick infestation (OR=1.5), poor milking hygiene (OR=2.4), and the presence of udder or teat lesions (OR=2.5) were also strongly associated with higher mastitis prevalence. In contrast, locality and body condition score showed no significant association. The study’s methodology included clinical examination, milk sampling, and structured questionnaires, with statistical analyses performed using SPSS. The authors emphasize the economic and public health burden of subclinical mastitis, which frequently escapes detection, and advocate for improved management practices, including hygiene protocols and tick control measures. The study received ethical approval from Benha University, declares no funding or competing interests, and confirms that data are available upon reasonable request. Overall, this research provides important epidemiological data that can inform targeted interventions to improve udder health and milk safety in Egyptian camel populations.

Review Comments and Suggestions

• Kindly present the prevalence of clinical and subclinical mastitis at both the camel level and the quarter level separately.

• Table 2:The title of column 2 suggests that dogs were sampled - kindly correct this.

• Line 158: Kindly italicise Camelus dromedarius.

Reviewer #2: The test/criteria for diagnosis of sub-clinical and clinical mastitis applied in the study is not adequate to get an authentic result. Furthermore, it is not clear that prevalence is quarterwise or animal wise.

**Do you want your identity to be public for this peer review?** For information about this choice, including consent withdrawal, please see our Privacy Policy

Reviewer #1: No

Reviewer #2: No

---

## [Author Response · Author response to Decision Letter 1]

7 Aug 2025

Comments to the Author

1. Is the manuscript technically sound, and do the data support the conclusions?

Reviewer #1: Yes

Reviewer #2: No

2. Has the statistical analysis been performed appropriately and rigorously?

Reviewer #1: Yes

Reviewer #2: Yes

3. Have the authors made all data underlying the findings in their manuscript fully available?

Reviewer #1: Yes

Reviewer #2: Yes

4. Is the manuscript presented in an intelligible fashion and written in standard English?

Reviewer #1: Yes

Reviewer #2: Yes

5. Review Comments to the Author

This research article, titled "Prevalence of Mastitis and Its Associated Risk Factors in Lactating Camels in Northern Egypt," published in PLOS ONE, presents a cross-sectional epidemiological investigation into camel mastitis across three governorates: Kafr El-Sheikh, Qalyubia, and Menofia. The study assessed 390 lactating camels using the California Mastitis Test (CMT) and identified both clinical and subclinical cases. The overall prevalence was 38.46%, with clinical mastitis accounting for 6.4% and subclinical mastitis for 32.1% of the examined animals. Multivariable logistic regression identified several significant risk factors: camels older than five years were 6.2 times more likely to be affected; early lactation stage increased risk by 2.5 times; tick infestation (OR=1.5), poor milking hygiene (OR=2.4), and the presence of udder or teat lesions (OR=2.5) were also strongly associated with higher mastitis prevalence. In contrast, locality and body condition score showed no significant association. The study’s methodology included clinical examination, milk sampling, and structured questionnaires, with statistical analyses performed using SPSS. The authors emphasize the economic and public health burden of subclinical mastitis, which frequently escapes detection, and advocate for improved management practices, including hygiene protocols and tick control measures. The study received ethical approval from Benha University, declares no funding or competing interests, and confirms that data are available upon reasonable request. Overall, this research provides important epidemiological data that can inform targeted interventions to improve udder health and milk safety in Egyptian camel populations.

Review Comments and Suggestions

Point#: Kindly present the prevalence of clinical and subclinical mastitis at both the camel level and the quarter level separately.

Response# it was clarified

Point#: Table 2:The title of column 2 suggests that dogs were sampled - kindly correct this.

Response# it corrected

Point#: Line 158: Kindly italicise Camelus dromedarius.

Response# it edited

Reviewer 2:

Comments on the manuscript entitled “Prevalence of Mastitis and Its Associated Risk Factors in Lactating Camels in Northern Egypt”

General Comments

This is an informative study, revealing various risk factors for sub-clinical and clinical mastitis in camel. However, a significant weakness is the use of CMT as sole tool for mastitis diagnosis

Point#: For accurate and sensitive subclinical mastitis diagnosis, use of somatic cell count and/or isolation of pathogenic bacteria from milk are also suggested to include.

The manuscript is well written and statistical analysis is also appropriate.

Materials & Methods

Point#: The sub-clinical and clinical mastitis in camel was diagnosed using CMT that was originally developed for use in cattle. Though an old technique, it still serves as backbone of mastitis diagnosis. However, several paper has suggested certain modifications for improving diagnostic sensitivity and specificity in camel, as chemical properties of normal as well as mastitic camel milk differ from those in cattle. Authors should describe in detail the scoring system. I am unable to conclude what is 3% CMT reagent?

Point#: Furthermore, I am unable to decide whether the prevalence is calculated as animal basis or udder-quarter basis. The authors should clarify how many quarters were examined and how much were found positive for mastitis? Results should be presented as both quarter wise and animal wise prevalence.

Response#: it was clarified and added

Point#: Criteria for body condition scoring is not described.

Response#: it added

Specific Comments

Point#: Line No. 39: Revise the sentence “It also contains higher phosphorus levels than other cattle”.

Response#: it deleted

---

## [Editor Report · Decision Letter 1]

19 Aug 2025

Dear Dr. Selim,

Thank you for submitting your manuscript to PLOS ONE. After careful consideration, we feel that it has merit but does not fully meet PLOS ONE’s publication criteria as it currently stands. Therefore, we invite you to submit a revised version of the manuscript that addresses the points raised during the review process.

In particular, I consider that you have only partially adressed the comments made by both reviewers on the original submission.

1- Table1 and table 2 are somewhat duplicated and should be compiled into a single table.

In such a table, you should indicate one column for « No of positive camels ^a^ », with its associated « Camel-level prevalence %» column, and a one column for « No of positive quarter samples ^b^ » and a « Quarter-level prevalence % » column. Subscripts a and b should indicate the total number of camels and quarters respectively as a footnote of the table.

2- In table 5, you should specify the distribution between clinical and subclinical cases. For instance, for each pathogen, you could mention in parenthesis the number of clinical and subclinical cases from which this pathogen was identified. If, as an example, among the 95 E. coli isolates 20 are from subclinical casees, you could indicate 95 (20/75).

3- Furthermore, you should clarify how you selected the 600 milk samples for bacteriological analysis since you only observed 450 mastitic quarters. For this last comment, I insist that you not only correct the manuscript file but also give details in your response to this comments. You will have to adjust the abstract.

We look forward to receiving your revised manuscript.

Kind regards,

Pierre Germon

Academic Editor

PLOS ONE
---

## [Author Response · Author response to Decision Letter 2]

1 Sep 2025

Point#: Table1 and table 2 are somewhat duplicated and should be compiled into a single table.

In such a table, you should indicate one column for « No of positive camels a», with its associated « Camel-level prevalence %» column, and a one column for « No of positive quarter samples b» and a « Quarter-level prevalence % » column. Subscripts a and b should indicate the total number of camels and quarters respectively as a footnote of the table.

Response#: both tables were merged

Point#: In table 5, you should specify the distribution between clinical and subclinical cases. For instance, for each pathogen, you could mention in parenthesis the number of clinical and subclinical cases from which this pathogen was identified. If, as an example, among the 95 E. coli isolates 20 are from subclinical casees, you could indicate 95 (20/75).

Response#: it rephrased

Point#: Furthermore, you should clarify how you selected the 600 milk samples for bacteriological analysis since you only observed 450 mastitic quarters. For this last comment, I insist that you not only correct the manuscript file but also give details in your response to this comments. You will have to adjust the abstract.

Response#: The results revealed that 150 animals tested positive. From these animals, 600 milk samples representing their quarters were subjected to bacteriological examination.

---

## [Editor Report · Decision Letter 2]

18 Sep 2025

Prevalence of Mastitis and Its Associated Risk Factors in Lactating Camels in Northern Egypt

PONE-D-25-32223R2

Dear Dr. Selim,

We’re pleased to inform you that your manuscript has been judged scientifically suitable for publication and will be formally accepted for publication once it meets all outstanding technical requirements.

Kind regards,

Pierre Germon

Academic Editor

PLOS ONE
---

## [Editor Report · Acceptance letter]

PONE-D-25-32223R2

PLOS ONE

Dear Dr. Selim,

I'm pleased to inform you that your manuscript has been deemed suitable for publication in PLOS ONE. Congratulations! Your manuscript is now being handed over to our production team.

Kind regards,

on behalf of

Dr. Pierre Germon

Academic Editor

PLOS ONE